# Adaptive Data Debiasing through Bounded Exploration

**Yifan Yang**
Ohio State University
yang.5483@osu.edu

**Yang Liu**
University of California, Santa Cruz
yangliu@ucsc.edu

**Parinaz Naghizadeh**
Ohio State University
naghizadeh.1@osu.edu

## Abstract

Biases in existing datasets used to train algorithmic decision rules can raise ethical and economic concerns due to the resulting disparate treatment of different groups. We propose an algorithm for sequentially debiasing such datasets through adaptive and bounded exploration in a classification problem with costly and censored feedback. Exploration in this context means that at times, and to a judiciously-chosen extent, the decision maker deviates from its (current) loss-minimizing rule, and instead accepts some individuals that would otherwise be rejected, so as to reduce statistical data biases. Our proposed algorithm includes parameters that can be used to balance between the ultimate goal of removing data biases – which will in turn lead to more accurate and fair decisions, and the exploration risks incurred to achieve this goal. We analytically show that such exploration can help debias data in certain distributions. We further investigate how fairness criteria can work in conjunction with our data debiasing algorithm. We illustrate the performance of our algorithm using experiments on synthetic and real-world datasets.

## 1 Introduction

Data-driven algorithmic decision making is being adopted widely to aid humans' decisions, in applications ranging from loan approvals to determining recidivism in courts. Despite their ability to process vast amounts of data and make accurate predictions, these algorithms can also exhibit and amplify existing social biases (e.g., [11, 23, 33]). There are at least two possible sources of unfairness in algorithmic decision rules: (data) biases in the training datasets, and (prediction) biases arising from the algorithm's decisions [29]. The latter problem has been receiving increasing attention, and is often addressed by imposing fairness constraints on the algorithm. In contrast, in this paper, we are primarily focused on the former problem of statistical biases in the training data itself.

The datasets used for training machine learning algorithms might not accurately represent the agents they make decisions on, due to, e.g., historical biases in decision making and feature selection, or changes in the populations' characteristics or participation rates since the data was initially collected. Such data biases in turn can result in disparate treatment of underrepresented or disadvantaged groups; i.e., data bias can cause prediction/model bias, as also verified by recent work [18, 40, 43, 25]. Motivated by this, we focus on data biases, and propose an algorithm which, while attempting to make accurate (and fair) decisions, also aims to collect data in a way that helps it recover unbiased estimates of the characteristics of agents interacting with it.

In particular, we study a classification problems with *censored and costly feedback*. Censored feedback means that the decision maker only observes the true qualification state of those individuals it admits (e.g., a bank will only observe whether an individual defaults on or repays a loan if the loan is extended in the first place; an employer only assesses the performance of applicants it hires). In such settings, any mismatch between the available training data and the true population may grow over time due to adaptive sampling bias: once a decision rule is adopted based on the current training

data, the algorithm's decisions will impact new data collected in the future, in that only agents passing the requirements set by the current decision rule will be admitted going forward. In response, the decision maker may attempt to collect more data from the population; however, such data collection is costly (e.g., in the previous examples, may require extending loans to/hiring unqualified individuals). Given these challenges, we present an *active debiasing* algorithm with *bounded exploration*: our algorithm admits some agents that would otherwise be rejected (i.e., it explores), yet adaptively and judiciously limits the extent and frequency of this exploration.

Formally, consider a population of agents with features $x$, true qualification/labels $y$, and group memberships $g$ based on their demographic features. To design a (fair) algorithm that can minimize classification loss, the decision maker (implicitly) relies on estimates $\hat{f}_{g,t}^y(x)$ of the feature-label distribution of agents from group $g$, obtained from the current training dataset $\mathcal{H}_t = \{(x_n, y_n, g_n)\}_{n=1}^{N_t}$. However, the resulting assumed distribution $\hat{f}_{g,t}^y(x)$ may be different from the true underlying distribution $f_g^y(x)$; this is the statistical data bias issue we focus on herein. Specifically, we consider distribution shifts between the estimates and the true distributions (Assumption 1).

**Our algorithm.** We propose an `active debiasing` algorithm (Algorithm 1), which actively adjusts its decisions with the goal of ensuring unbiased estimates of the underlying distributions $f_g^y(x)$ over time. In particular, at each time $t$, the algorithm selects a (fairness-constrained) decision rule that would minimize classification error based on its current, possibly biased estimates $\hat{f}_{g,t}^y(x)$; adopting this decision rule corresponds to *exploitation* of the current information by the algorithm. At the same time, to circumvent the censored feedback nature of the problem, our algorithm also deviates from the prescriptions of this loss-minimizing classifier to a judiciously chosen extent (the extent is chosen adaptively, based on the current estimates); this will constitute *exploration*. Our algorithm includes two parameters to limit the costs of this exploration: one modulates the *frequency* of exploration (an exploration probability $\epsilon_t$ which can be adjusted using current bias estimates), and another limits the *depth* of exploration (by setting a threshold $\text{LB}_t$ on how far from the classifier one is willing to go when exploring). We show that these choices can strike a balance between the ultimate goal of removing statistical biases in the training data – which will in turn lead to more accurate and fair decisions, and the cost of exploration incurred to achieve this goal.

**Summary of findings and contributions.** Our main findings and contributions are as follows:

1. *Comparison with baselines.* We contrast our proposed algorithm against two baselines: an `exploitation-only` baseline (one that does not include any form of exploration), and a `pure exploration` baseline (which may randomly accept some of the agents rejected by the classifier, but does not bound exploration). We show (Theorem 1) that `exploitation-only` always leads to overestimates of the underlying distributions. Further, while `pure exploration` can debias the distribution estimates in the long-run (Theorem 2), it does so at the expense of accepting *any* agent, no matter how far from the classifier's threshold, leading to more costly exploration (Section 5).

2. *Analytical support for our proposed algorithm.* We show (Theorem 3) that our proposed `active debiasing` algorithm with bounded exploration can correct biases in unimodal distribution estimates. We also provide an error bound for our algorithm (Theorem 4).

3. *Interplay with fairness criteria.* We analyze the impact of fairness constraints on our algorithm's performance, and show (Proposition 1) that existing fairness criteria may speed up debiasing of the data in one group, while slowing it down for another.

4. *Numerical experiments.* We provide numerical support for the performance of our algorithm using experiments on synthetic and real-world (Adult and FICO) datasets.

**Related work.** Our paper is closely related to the works of [4, 20, 14, 6, 17], which study the impact of data biases on (fair) algorithmic decision making. Among these works, Bechavod et al. [4] and Kilbertus et al. [20] study *fairness-constrained* learning in the presence of censored feedback. While these works also use exploration, the form and purpose of exploration is different: the algorithm in [4] starts with a pure exploration phase, and subsequently explores with the goal of ensuring the fairness constraint is not violated; the stochastic (or exploring) policies in [20] conduct (pure) exploration to address the censored feedback issue. In contrast, we start with a biased dataset, and conduct *bounded* exploration to debias data; fairness constraints may or may not be enforced separately and are orthogonal to our debiasing process. Also, as shown in Section 5, such pure exploration processes can incur higher exploration costs than our proposed bounded exploration algorithm.

Our work is also closely related to [10, 31, 30, 41], which study adaptive sampling biases induced by a decision rule, particularly when feedback is censored. Among these, Neel and Roth [30] also consider an *adaptive* data gathering procedure, and show that no debiasing will be necessary if the data is collected through a differentially private method. We similarly propose an adaptive debiasing algorithm, but unlike [30], account for the costs of exploration in our data collection procedure. The recent work of Wei [41] studies data collection in the presence of censored feedback, and similar to our work, accounts for the cost of exploration in data collection, by formulating the problem as a partially observable Markov decision processes. Using dynamic programming methods, the data collection policy is shown to be a threshold policy that becomes more stringent (in our terminology, reduces exploration) as learning progresses. Our works are similar in that we both propose using adaptive and cost-sensitive exploration, but we differ in the problem setup and our analysis of the impact of fairness constraints. More importantly, in contrast to both [30, 41], our starting point is a *biased* dataset (which may be biased for reasons other than adaptive sampling in its collection, including historical biases); we then show how, while attempting to debias this dataset by collecting new data, any additional adaptive sampling bias during data collection can be prevented.

Our work also falls within the fields of selective labeling bias, fair learning, and active learning. From the *selective labeling bias* perspective, Lakkaraju et al. [22] propose a contraction technique to compare the performance of the predictive model and a human judge while they are forced to have the same acceptance rate. De-Arteaga et al. [9] propose a data augmentation scheme by adding more samples that are more likely to be rejected (we refer to this as exploration) to correct the sample selection bias. From the *fair learning* perspective, Kallus and Zhou [18] propose a re-weighting technique (re-weighting ideas are also explored in [1, 6, 17]) to solve the residual unfairness issue while accounting for adaptive sampling bias. From the *active learning* perspective, Noriega-Campero et al. [32] adaptively acquire additional information according to the needs of different groups or individuals given information budgets, to achieve fair classification. Similar to the approaches of these papers, we also compensate for adaptive sampling bias through exploration; the main difference, aside from the application, is in our analytical guarantees as well as our study of the interplay of data debiasing with fairness constraints.

More broadly, our work has similarities to Bandit learning and its focus on exploration-exploitation trade-offs. A key difference of our work with existing bandit algorithms ($\epsilon$-greedy, UCB, EXP3, etc.) is our focus on *bounded* exploration. We provide additional discussion on this, and review other related works [35, 3, 19, 26, 27, 42, 22, 9, 18, 32, 1] in more detail, in Appendix B.

## 2 Model and Preliminaries

**The environment.** We consider a *firm* or decision maker, who selects an algorithm to make decisions on a population of *agents*. The firm observes agents arriving over times $t = 1, 2, \ldots$, makes a decision for agents arriving at time $t$ based on the current algorithm, and can subsequently adjust its algorithm for times $t + 1$ onward based on the observed outcomes.

Each agent has an observable *feature* or *score* $x \in \mathcal{X} \subseteq \mathbb{R}$.[1] These represent the agent characteristics that are leveraged by the firm in its decision; examples include credit scores or exam scores. Each agent is either qualified or unqualified to receive a favorable decision; this is captured by the agent's true *label* or *qualification state* $y \in \{0, 1\}$, with $y = 1$ and $y = 0$ denoting qualified and unqualified agents, respectively. In addition, each agent in the population belongs to a different *group* based on its demographic or protected attributes (e.g., race, gender); the agent's group membership is denoted $g \in \{a, b\}$. We consider threshold-based, group-specific, binary classifiers $h_{\theta_{g,t}}(x) : \mathcal{X} \to \{0, 1\}$ as (part of) the algorithm adopted by the firm, where $\theta_{g,t}$ denotes the classifier's decision threshold. An agent from group $g$ with feature $x$ arriving at time $t$ is admitted if $x \geq \theta_{g,t}$.

**Quantifying bias.** Let $f_g^y(x)$ denote the true underlying probability density function for the feature distribution of agents from group $g$ with qualification state $y$. The algorithm has an estimate of these unknown distributions, at each time $t$, based on the data collected so far (or an initial training set). Denote the algorithm's estimate at $t$ by $\hat{f}_{g,t}^y(x)$. In general, there can be a mismatch between the estimates $\hat{f}_{g,t}^y(x)$ and the true $f_g^y(x)$; this is what we refer to as bias. We assume the following.

---

[1]We use a one-dimensional feature setting in our analysis, and generalize to $\mathcal{X} \subseteq \mathbb{R}^n$ in Section 5. Discussions and numerical experiments on potential loss of information due to our feature dimension reduction technique is given in Appendix A.

**Assumption 1.** *The firm updates its estimates $\hat{f}_{g,t}^y(x)$ by updating a single parameter $\hat{\omega}_{g,t}^y$.*

This type of assumption is common in the multi-armed bandit learning literature [38, 39, 34, 24, 36] (there, the algorithm aims to learn the mean arm rewards). In our setting, it holds when the assumed underlying distribution is single-parameter, or when only one of the parameters of a multi-parameter distribution is unknown. Alternatively, it can be interpreted as identifying and correcting distribution shifts by updating a reference point in the distribution (e.g., adjusting the mean).[2] More specifically, we will let $\hat{\omega}_{g,t}^y$ be the $\alpha$-th percentile of $\hat{f}_{g,t}^y(x)$. We discuss potential limitations of Assumption 1 in Appendix A, and present an extension to a case with two unknown parameters in Appendix I.

Under Assumption 1, the bias can be captured by the mismatch between the estimated and true parameters $\hat{\omega}_{g,t}^y$ and $\omega_g^y$. In particular, we set the mean absolute error $\mathbb{E}[|\hat{\omega}_{g,t}^y - \omega_g^y|]$ as the measure for quantifying bias, where the randomness is due to that in $\hat{\omega}_{g,t}^y$, the estimate of the unknown parameter based on data collected up to time $t$.

**Algorithm choice without debiasing.** Let $\alpha_g^y$ be the fraction of group $g$ agents with label $y$. A loss-minimizing fair algorithm selects its thresholds $\theta_{g,t}$ at time $t$ as follows:

$$\min_{\theta_{a,t}, \theta_{b,t}} \quad \sum_{g \in \{a,b\}} \alpha_g^1 \int_{-\infty}^{\theta_{g,t}} \hat{f}_{g,t}^1(x)\mathrm{d}x + \alpha_g^0 \int_{\theta_{g,t}}^{\infty} \hat{f}_{g,t}^0(x)\mathrm{d}x, \quad \text{s.t.} \quad \mathcal{C}(\theta_{a,t}, \theta_{b,t}) = 0 \,. \tag{1}$$

Here, the objective is the misclassification error, and $\mathcal{C}(\theta_a, \theta_b) = 0$ is the fairness constraint imposed by the firm, if any. For instance, $\mathcal{C}(\theta_{a,t}, \theta_{b,t}) = \theta_{a,t} - \theta_{b,t}$ for *same decision rule*, or $\mathcal{C}(\theta_{a,t}, \theta_{b,t}) = \int_{\theta_{a,t}}^{\infty} \hat{f}_{a,t}^1(x)\mathrm{d}x - \int_{\theta_{b,t}}^{\infty} \hat{f}_{b,t}^1(x)\mathrm{d}x$ for *equality of opportunity*. Note that both the objective function and the fairness constraint are affected by any inaccuracies in the current estimates $\hat{f}_{g,t}^y$. As such, a biased training dataset can lead to both loss of accuracy and loss in desired fairness.

# 3 An `Active Debiasing` Algorithm with Bounded Exploration

In this section, we present the `active debiasing` algorithm which uses both *exploitation* (the decision rules of (1)) and *exploration* (some deviations) to remove any biases from the estimates $\hat{f}_{g,t}^y$. Although the deviations may lead to admission of some unqualified agents, they can be beneficial to the firm in the long-run: by reducing biases in $\hat{f}_{g,t}^y$, both classification loss estimates and fairness constraint evaluations can be improved. In this section, we drop the subscripts $g$ from the notation; when there are multiple groups, our algorithm can be applied to each group's estimates separately.

As noted in Section 1, our algorithm is one of *bounded* exploration: it includes a *lower bound* $LB_t$, which captures the extent to which the decision maker is willing to deviate from the current classifier $\theta_t$, based on its current estimate $\hat{f}_t^0$ of the unqualified agents' underlying distribution. Formally,

**Definition 1.** *At time $t$, the firm selects a lower bound $LB_t$ such that*

$$LB_t = (\hat{F}_t^0)^{-1}(2\hat{F}_t^0(\hat{\omega}_t^0) - \hat{F}_t^0(\theta_t)),$$

*where $\theta_t$ is the (current) loss-minimizing threshold determined from (1), $\hat{F}_t^0$, $(\hat{F}_t^0)^{-1}$ are the cdf and inverse cdf of the estimated distribution $\hat{f}_t^0$, respectively, and $\hat{\omega}_t^0$ is (wlog) the $\alpha$-th percentile of $\hat{f}_t^0$.*

In more detail, we choose $LB_t$ such that $\hat{F}_t^0(\hat{\omega}_t^0) - \hat{F}_t^0(LB_t) = \hat{F}_t^0(\theta_t) - \hat{F}_t^0(\hat{\omega}_t^0)$; that is, such that $\hat{\omega}_t^0$ is the median in the interval $(LB_t, \theta_t)$ based on the current estimate of the distribution $\hat{F}_t^0$ at the beginning of time $t$. Then, once a new batch of data is collected, we update $\hat{\omega}_t^0$ to $\hat{\omega}_{t+1}^0$, the *realized* median of the distribution between $(LB_t, \theta_t)$ based on the data observed during $[t, t+1)$. Once the underlying distribution is correctly estimated, (in expectation) we will observe the same number of samples between $(LB_t, \omega_t^0)$ and between $(\omega_t^0, \theta_t)$, and hence $\omega_t^0$ will no longer change. We also note that by selecting a high $\alpha$-th percentile in the above definition, $LB_t$ can be increased so as to limit the depth of exploration. As shown in Theorem 3, and in our numerical experiments, these thresholding choice will enable debiasing of the distribution estimates while controlling its costs.

Our `active debiasing` algorithm is summarized below. A pseudo-code is given in Appendix C.

---

[2]For instance, a bank may want to adjust for increases in average credit scores [15, 7] over time.

**Algorithm 1** (The `active debiasing` algorithm)**.** *Denote the loss-minimizing decision threshold determined from* (1) *by* $\theta_t$, *and let* $LB_t$ *be given by Definition 1. Let* $\{\epsilon_t\}$ *be a sequence of exploration probabilities. At each time* $t$, *and for agents* $(x^\dagger, y^\dagger)$ *arriving at* $t$:
***Step I: Admit agents and collect data.*** *Admit all agents with* $x^\dagger \geq \theta_t$. *Additionally, if* $LB_t \leq x^\dagger < \theta_t$, *admit the agent with probability* $\epsilon_t$.
***Step II: Update the distribution estimates based on new data collected in Step I.***
- ***Qualified agents' distribution update:*** *Identify new data with* $LB_t \leq x^\dagger$ *and* $y^\dagger = 1$. *Use all such* $x^\dagger$ *with* $LB_t \leq x^\dagger < \theta_t$, *and such* $x^\dagger$ *with* $\theta_t \leq x^\dagger$ *with probability* $\epsilon_t$, *to update* $\hat{\omega}_t^1$.
- ***Unqualified agents' distribution update:*** *Identify new data with* $LB_t \leq x^\dagger$ *and* $y^\dagger = 0$. *Use all such* $x^\dagger$ *with* $LB_t \leq x^\dagger < \theta_t$, *and such* $x^\dagger$ *with* $\theta_t \leq x^\dagger$ *with probability* $\epsilon_t$, *to update* $\hat{\omega}_t^0$.

In more detail, our algorithm repeatedly performs the following two steps:

**Step I: Data collection.** At the beginning of a time period $t$, a loss-minimizing classifier with threshold $\theta_t$ (according to (1)) and the exploration lower bound $LB_t$ (Definition 1) are selected based on the data collected so far. Then, given $\theta_t$, the new data collected during period $t$ will consist of arriving agents with features $x \geq \theta_t$. Additionally, to address the censored feedback issues, with probability $\epsilon_t$, the algorithm will also accept agents with $LB_t \leq x < \theta_t$. Note that this step balances between exploration and exploitation through its choice of both $LB_t$ (which limit the depth of exploration) and exploration probabilities $\epsilon_t$ (which limits the frequency of exploration).

**Step II: Updating estimates.** At the end of period $t$, the data collected in Step I will be used to update $\hat{f}_t^0$ and $\hat{f}_t^1$. Under Assumption 1, the estimates $\hat{f}_t^y$ are updated by updating the parameter $\hat{\omega}_t^y$. We assume, without loss of generality, that the firm sets $\hat{\omega}_t^y$ to the $\alpha$-th percentile of $\hat{f}_t^y$. This $\alpha$-th percentile is the *reference point* that will be adjusted over time as new data is collected. As an example, when the reference point $\hat{\omega}_t^1$ is set to the median (the 50-th percentile), the parameter can be adjusted so that half the label 1 data collected in Step I will lie on each side of the reference point.

## 4 Theoretical Analysis

We begin by analyzing two baselines: `exploitation-only` (which only accepts agents with $x \geq \theta_t$, and uses no exploration or thresholding) and `pure exploration` (which accepts arriving agents at time $t$ who have $x < \theta_t$ with probability $\epsilon_t$, without setting any lower bound). The motivation for the choice of these two baselines is as follows: the `exploitation-only` baseline tracks the performance of a decision maker who is unaware of underlying data biases, and makes no attempt at fixing them. The `pure exploration` baseline, on the other hand, is motivated by the Bandit learning literature, and is also akin to debiasing algorithms proposed in recent work (see Section 1, Related Work). We, in contrast, propose and show the benefits of bounded exploration through our `active debiasing` algorithm.

### 4.1 The exploitation-only baseline

Our first baseline algorithm only updates its estimates of the underlying distributions based on agents with $x \geq \theta_t$ who pass the (current) loss-minimizing classifier (1). The following result shows that this approach consistently suffers from adaptive sampling bias, ultimately resulting in overestimation of the underlying distributions.

**Theorem 1.** *An `exploitation-only` algorithm overestimates* $\omega^y$, *i.e.,* $\lim_{t \to \infty} \mathbb{E}[\hat{\omega}_t^y] > \omega^y, \forall y$.

A detailed proof is given in Appendix D.

### 4.2 The pure exploration baseline

In this second baseline, at each time $t$, the algorithm may accept any agent with $x < \theta_t$ with probability $\epsilon_t$. The following result establishes that using the data collected this way, the distributions can be debiased in the long-run, *if* the data collected above the classifier is also sampled with probability $\epsilon_t$ when updating the distributions.

**Theorem 2.** *Using the `pure exploration` algorithm,* $\hat{\omega}_t^y \to \omega^y$ *as* $t \to \infty$, $\forall y$.

The proof follows from assuming (wlog) that the unknown parameter $\omega^y$ being estimated is the distribution's mean (can be generalized to arbitrary statistics under Assumption 1). Then, as we are collecting i.i.d. samples from across the distribution, $\hat{\omega}_t^y$ can be set to the sample mean of the collected data, and the conclusion follows from the strong law of large numbers. Note also that if *all* the data above the classifier was considered when making the updates, following similar arguments to those in the proof of Theorem 1, the algorithm would obtain overestimates of the distributions. Lastly, we could equivalently balance data by resampling the exploration data (rather than downsampling the exploitation data), to debias data through this procedure.

### 4.3 The `active debiasing` **algorithm**

While `pure exploration` can successfully debias data in the long-run, it does so at the expense of accepting agents with *any* $x < \theta_t$. Below, we provide analytical support that our proposed exploration and thresholding procedure in the `active debiasing` algorithm can still debias data in certain distributions, while limiting the depth of exploration to $\text{LB}_t < x < \theta_t$.

**Theorem 3.** *Let $f^y$ and $\hat{f}_t^y$ denote the true feature distribution and their estimates at the beginning of time $t$, with respective $\alpha$-th percentiles $\omega^y$ and $\hat{\omega}_t^y$. Assume these are unimodal distributions, $\epsilon_t > 0, \forall t$, and $\hat{\omega}_t^0 \leq \theta_t \leq \hat{\omega}_t^1, \forall t$. Then, using the `active debiasing` algorithm,*
*(a) If $\hat{\omega}_t^y$ is underestimated (resp. overestimated), then $\mathbb{E}[\hat{\omega}_{t+1}^y] \geq \hat{\omega}_t^y$, (resp. $\mathbb{E}[\hat{\omega}_{t+1}^y] \leq \hat{\omega}_t^y$) $\forall t, \forall y$.*
*(b) The sequence $\{\hat{\omega}_t^y\}$ converges, with $\hat{\omega}_t^y \to \omega^y$ as $t \to \infty$, $\forall y$.*

We provide a proof sketch for debiasing $\hat{f}_t^0$ which highlights the main technical challenges addressed in our analysis. The detailed proof is given in Appendix E.

*Proof sketch:* Our proof involves the analysis of statistical estimates $\hat{\omega}_t^0$ based on data collected from *truncated* distributions. In particular, by bounding exploration, our algorithm will only collect data with features $x \geq \text{LB}_t$, and can use only this truncated data to build estimates of the unknown parameter of the distributions.

Part **(a)** establishes that the sequence of $\{\hat{\omega}_t^y\}$ produced by our `active debiasing` algorithm "moves" in the right direction over time, and ultimately converges. The main challenge in this analysis is that as the exploration and update intervals $[\text{LB}_t, \infty)$ are themselves adaptive, there is no guarantee on the number of samples in each interval, and therefore we need to analyze the estimates in finite sample regimes. To proceed with the analysis, we assume the feature distribution estimates follow unimodel distributions (such as Gaussian, Beta, and the family of $alpha$-stable distributions) with $\omega^0$ as reference points. We then consider the expected parameter update following the arrival of a batch of agents; Denote the current left portion in $(\text{LB}_t, \hat{\omega}_t^0)$ as $p_1 := \frac{\hat{F}^0(\hat{\omega}_t^0) - \hat{F}^0(\text{LB}_t)}{\hat{F}^0(\theta_t) - \hat{F}^0(\text{LB}_t)}$. Based on Definition 1, we can also obtain the current portion in $(\hat{\omega}_t^0, \theta_t)$ denoted as $p_2 := \frac{\hat{F}^0(\theta_t) - \hat{F}^0(\hat{\omega}_t^0)}{\hat{F}^0(\theta_t) - \hat{F}^0(\text{LB}_t)} = p_1$. The new expected estimates $\mathbb{E}[\hat{\omega}_{t+1}^0]$ is the sample median in $(\text{LB}_t, \theta_t)$, where samples come from the true distribution. We establish that this expected update will be higher/lower than $\omega_t^0$ if the current estimate is an under/over estimate of the true parameter.

Then, in Part **(b)** we first show that the sequence of over- and under-estimation errors in $\{\hat{\omega}_t^y\}$ relative to the true parameter $\omega^y$ are supermartingales. By the Doobs Convergence theorem and using results from part **(a)**, these will converge to zero mean random variables with variance going to zero as the number of samples increases. This establishes that $\{\hat{\omega}_t^y\}$ converges. It remains to show that this convergence point is the true parameter of the distribution. To do so, as detailed in the proof, we note that the density function of the sample median estimated on label 0 data collected in $[\text{LB}_t, \theta_t]$ is

$$\mathbb{P}(\hat{\omega}_t^0 = \nu)\mathrm{d}\nu = \frac{(2m+1)!}{m!m!}\left(\frac{F^0(\nu) - F^0(\text{LB}_t)}{F^0(\theta_t) - F^0(\text{LB}_t)}\right)^m \left(\frac{F^0(\theta_t) - F^0(\nu)}{F^0(\theta_t) - F^0(\text{LB}_t)}\right)^m \frac{f^0(\nu)}{F^0(\theta_t) - F^0(\text{LB}_t)}\mathrm{d}\nu \quad (2)$$

which is a beta distribution pushed forward by $H(\nu) := \frac{F^0(\nu) - F^0(\text{LB}_t)}{F^0(\theta_t) - F^0(\text{LB}_t)}$; this is the CDF of the truncated $F^0$ distribution in $[\text{LB}_t, \theta_t]$. We then establish that the convergence point will be the true median of the underlying distribution. $\square$

### 4.4 Error bound analysis

Our error bound analysis compares the errors (measured as the number of wrong decisions made) of our `adaptive debiasing` algorithm against the errors that would be made by an oracle which knows

the true underlying distributions. We measure the performance using 0-1 loss, $\ell(\hat{y}_i, y_i) = \mathbb{1}[\hat{y}_i \neq y_i]$, where $\hat{y}_i$ and $y_i$ denote the predicted and true label of agent $i$, respectively. We consider the error accumulated when updating the estimates using a total of $m$ batches of data. We split the total $T$ samples that have arrived during $[t, t+1)$ into four groups, corresponding to four different distributions $f_g^y$. Specifically, we use $b_{g,t}^y$ to denote the number of samples from each label-group pair at round $t \in \{0, \ldots, m\}$. We update the unknown distribution estimates once all batches meet a size requirement $s$, i.e, once $\min(b_{g,t}^y) \geq s, \forall y, \forall g$. The error of our algorithm is given by:

$$\text{Error} = \mathbb{E}[Error_{Adaptive} - Error_{Oracle}]$$

$$= \sum_t \sum_{i=1}^{b_{a,t}^0 + b_{a,t}^1 + b_{b,t}^0 + b_{b,t}^1} \mathbb{E}_{(x_i, y_i, g_i) \sim D} \left[ \ell(h_{\theta_{t,g}}(x_i, g_i), y_i) \right] - \sum_{i=1}^T \mathbb{E}_{(x_i, y_i, g_i) \sim D} \left[ \ell(h_{\theta_g^*}(x_i, g_i), y_i) \right]$$

The following theorem provides an upper bound on the error incurred by `active debiasing`.

**Theorem 4.** *Let $\hat{f}_{g,t}^y(x)$ be the estimated feature-label distributions at round $t \in \{0, \ldots, m\}$. We consider the threshold-based, group-specific, binary classifier $h_{\theta_{g,t}}$, and denote the Rademacher complexity of the classifier family $\mathcal{H}$ with $n$ training samples by $\mathcal{R}_n(\mathcal{H})$. Let $\theta_{g,t}$ be a $v$-approximately optimal classifier based on data collected up to time $t$. At round $t$, let $N_{g,t}$ be the number of exploration errors incurred by our algorithm, $n_{g,t}$ be the sample size at time $t$ from group $g$, $d_{\mathcal{H} \Delta \mathcal{H}}(\tilde{D}_{g,t}, D_g)$ be the distance between the true unbiased data distribution $D_g$ and the current biased estimate $\tilde{D}_{g,t}$, and $c(\tilde{D}_{g,t}, D_g)$ be the minimum error on an algorithm trained on unbiased and biased data. Then, with probability at least $1 - 4\delta$ with $\delta > 0$, the active debiasing algorithm's error is bounded by:*

$$Err. \leq \sum_{g,t} \Big[ \underbrace{2v}_{v\text{-approx.}} + \underbrace{4\mathcal{R}_{n_{g,t}}(\mathcal{H}) + \frac{4}{\sqrt{n_{g,t}}} + \sqrt{\frac{2 \ln(2/\delta)}{n_{g,t}}}}_{\text{empirical estimation errors}} + \underbrace{N_{g,t}}_{\text{explor.}} + \underbrace{d_{\mathcal{H} \Delta \mathcal{H}}(\tilde{D}_{g,t}, D_g) + 2c(\tilde{D}_{g,t}, D_g)}_{\text{source-target distribution mismatch}} \Big]$$

More details on the definitions of the distance measure $d_{\mathcal{H} \Delta \mathcal{H}}$, and the error term $c(\cdot)$, and the exploration error term $N_{g,t}$, along with a a detailed proof, are given in Appendix F. From the expression above, we can see that the error incurred by our algorithm consists of four types of error: errors due to approximation of the optimal (fair) classifier at each round, empirical estimation errors, exploration errors, and errors due to our biased training data (viewed as source-target distribution mismatches); the latter two are specific to our `active debiasing` algorithm. In particular, as we collect more samples, $n_{g,t}$ will increase. Hence, the empirical estimation errors decrease over time. Moreover, as the mismatch between $\tilde{D}_{g,t}$ and $D_g$ decreases using our algorithm (by Theorem 3), the error due to target domain and source domain mismatches also decrease. In the meantime, our exploration probability $\epsilon_t$ also becomes smaller over time, decreasing $N_{g,t}$.

### 4.5 Active debiasing and fairness criteria

We next consider our proposed `active debiasing` algorithm when used in conjunction with demographic fairness constraints (e.g., equality of opportunity, same decision rule, and statistical parity [29]). Imposing such fairness rules will lead to changes to the selected classifiers compared to the fairness-unconstrained case. Let $\theta_{g,t}^F$ and $\theta_{g,t}^U$ denote the fairness constrained and unconstrained decision rules obtained from (1) at time $t$ for group $g$, respectively. We say group $g$ is being over-selected (resp. under-selected) following the introduction of fairness constraints if $\theta_{g,t}^F < \theta_{g,t}^U$ (resp. $\theta_{g,t}^F > \theta_{g,t}^U$). Below, we show how such over/under-selections can differently affect the debiasing of estimates on different agents.

In particular, let the speed of debiasing be the rate at which $\mathbb{E}[|\hat{\omega}_t^y - \omega^y|]$ decreases with respect to $t$; then, for a given $t$, an algorithm for which this error is larger has a slower speed of debiasing. The following proposition identifies the impacts of different fairness constraints on the speed of debiasing attained by our `active debiasing` algorithm. The proof is given in Appendix G.

**Proposition 1.** *Let $f_g^y$ and $\hat{f}_{g,t}^y$ be the true and estimated feature distributions, with respective $\alpha$-th percentiles $\omega^y$ and $\hat{\omega}_t^y$. Assume these are unimodal distributions, and `active debiasing` is applied. If group $g$ is over-selected (resp. under-selected) under a fairness constraint, i.e., $\theta_{g,t}^F < \theta_{g,t}^U$ (resp. $\theta_{g,t}^F > \theta_{g,t}^U$), the speed of debiasing on the estimates $\hat{f}_{g,t}^y$ will decrease (resp. increase).*

Proposition 1 highlights the following implications of using both fairness rules and our active debiasing efforts. Some fairness constraints (such as equality of opportunity) can lead to an increase in opportunities for (here, over-selection of) agents from disadvantaged groups, while others (such as same decision rule) can lead to under-selection from that group. Proposition 1 shows that `active debiasing` may in turn become faster or slower at debiasing estimates on this group.

Intuitively, over-selection provides increased opportunities to agents from a group (compared to an unconstrained classifier). In fact, the reduction of the decision threshold to $\theta_{g,t}^F$ can itself be interpreted as introducing exploration (which is separate from that introduced by our debiasing algorithm). When a group is over-selected under a fairness constraint, the fairness-constrained threshold $\theta_{g,t}^F$ will be lower than the unconstrained threshold $\theta_{g,t}^U$. Therefore, the exploration range will be narrower, which means by adding a fairness constraint, the algorithm needs to wait and collect more samples (takes a longer time) before it manages to collect sufficient data to accurately update the unknown distribution parameter, and hence, it has a slower debiasing speed. More broadly, these findings contribute to our understanding of how fairness constraints can have long-term implications beyond the commonly studied fairness-accuracy tradeoff when we consider their impacts on data collection and debiasing efforts.

## 5 Numerical Experiments

In this section, we illustrate the performance of our algorithm through numerical experiments on both Gaussian and Beta distributed synthetic datasets, and on two real-world datasets: the *Adult* dataset [12] and the *FICO* credit score dataset [37] pre-processed by [16]. Additional details (ground-truth information) on the experiments, and larger versions of all figures, are available in Appendix H. Our code is available at: https://github.com/Yifankevin/adaptive_data_debiasing.

Throughout, we either choose a *fixed* schedule for reducing the exploration frequencies $\{\epsilon_t\}$, or reduce these *adaptively* as a function of the estimated error. For the latter, the algorithm can select a range (e.g., above the classifier for label 0/1) and adjust the exploration frequency proportional to the discrepancy between the number of observed classification errors in this interval relative to the number expected given the distribution estimates.

**Comparison with the `exploitation-only` and `pure exploration` baselines:** Our first experiments in Fig. 1, compare our algorithm against two baselines. The underlying distributions are Gaussian and no fairness constraint is imposed. Our algorithm sets $\alpha^1 = 50$ and $\alpha^0 = 60$ percentiles, and exploration frequencies $\epsilon_t$ are selected adaptively by both our algorithm and pure exploration.

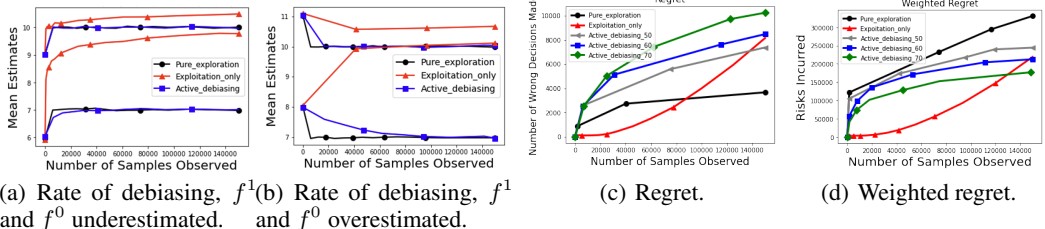

(a) Rate of debiasing, $f^1$ and $f^0$ underestimated.
(b) Rate of debiasing, $f^1$ and $f^0$ overestimated.
(c) Regret.
(d) Weighted regret.

Figure 1: Speed of debiasing, regret, and weighted regret, of `active debiasing` vs. `exploitation-only` and `pure exploration` (larger figures in Appendix H).

Speed of debiasing: Figs. 1(a) and 1(b) show that consistent with Theorem 1, `exploitation-only` overestimates the distributions due to adaptive sampling biases. Further, consistent with Theorem 2, `pure exploration` successfully debiases data. We also observe that as expected, `pure exploration` debiases *faster* than `active debiasing`. The difference is more pronounced in the label 0 distributions compared to label 1, where `pure exploration` collects more "diverse" observations than our algorithm. For this same reason, the gap between `pure exploration` and our algorithm is larger when $f^0$ is overestimated. This is because `pure exploration` observes samples with *lower* features $x$ than `active debiasing`, and so can use these to reduce its estimate faster.

Regret: Figs. 1(c) and 1(d) compare the regret and weighted regret of the algorithms. Regret is measured as the difference between the number of FN+FP decisions of an algorithm vs the oracle

loss-minimizing algorithm derived on unbiased data. Formally, regret is defined as in Section 4.4; weighted regret is defined similarly, but also adds a weight to each FN or FP decision, with the weight exponential in the distance of the feature of the admitted agent from the classifier. We observe that `exploitation-only`'s regret is super-linear, as not only it fails to debias, but has increasing error due to biases from overestimating. On the other hand, while algorithms that explore "deeper" have lower regret (e.g. `pure exploration` < `active debiasing` with $\alpha^0 = 50$ < `active debiasing` with $\alpha^0 = 60$ in Figs. 1(c)), they have higher weighted regret (the order is reversed in Fig. 1(d)). In other words, exploring to admit agents with low features $x$ leads to some errors, but ultimately helps reduce future mistakes, leading to sub-linear regret. However, if the risk/cost of these wrong decisions is taken into account, the firm may be better off adopting slower, but less risky exploration thresholds (e.g. $\alpha^0 = 70$).

**Performance of `active debiasing` on Beta distributions:** Fig. 2 shows that our algorithm can debias data for which the underlying feature-label distributions follow Beta distributions. We have assumed a mistmach between the parameter $\alpha$ of the true and estimated distributions, and selected these so that the estimated and true distributions have different relative skewness. This verifies that Theorem 3 holds for asymmetric distributions.

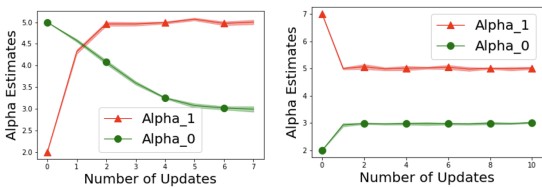

Figure 2: `Debiasing` under Beta distributions.

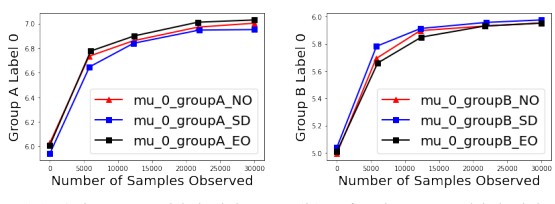

(a) Advantaged label 0.    (b) Disadvantaged label 0.

Figure 3: `Debiasing` used with fairness constraints.

**Interplay of debiasing and fairness constraints:** Fig. 3 compares the performance of `active debiasing` when there are two groups of agents with underlying Gaussian distributions, and the algorithm is chosen subject to three different fairness settings: no fairness, equality of opportunity (EO), and the same decision rule (SD). The findings are consistent with Proposition 1. For instance, SD will over-select the majority group (i.e., $\theta_{a,t}^{SD} < \theta_{a,t}^{U}$) so that, as shown in the left panel in Fig. 3, the speed of debiasing on the estimates $\hat{f}_{a,t}^y$ will decrease. In contrast, an opposite effect will happen in the minority group $b$ which is under-selected (i.e., $\theta_{b,t}^{SD} > \theta_{b,t}^{U}$). The effects of EO can be similarly explained by noting that it under-selects the majority group and over-selects the minority group.

`Active debiasing` **on the *Adult* dataset:** Fig. 4 illustrates the performance of our algorithm on the *Adult* dataset. Data is grouped based on race (White $G_a$ and non-White $G_b$), with labels $y = 1$ for income $> \$50k/year$. A one-dimensional feature $x \in \mathbb{R}$ is constructed by conducting logistic regression on four quantitative and qualitative features (education number, sex, age, workclass), based on the initial training data.[3] Using an input analyzer, we found Beta distributions as the best fit to the underlying distributions. We use 2.5% of the data to obtain a biased estimate of the parameter $\alpha$. The remaining data arrives sequentially. We use $\alpha^1 = 50$ and $\alpha^0 = 60$ and a fixed decreasing $\{\epsilon_t\}$, with the equality of opportunity fairness constraint imposed throughout.

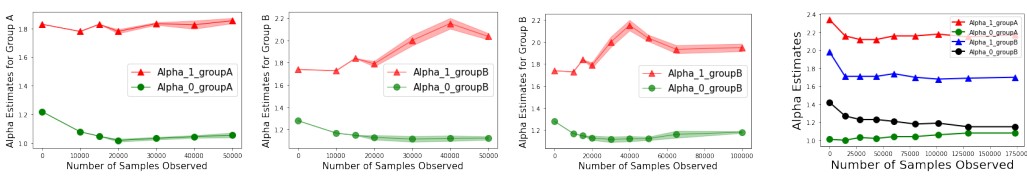

(a) Debiasing $G_a$, Adult.    (b) Debiasing $G_b$, Adult.    (c) $G_b$, augmented with synthetic data (Adult).    (d) `Debiasing` on *FICO*.

Figure 4: `Active debiasing` on the *Adult* and *FICO* datasets.

We observe that our proposed algorithm can debias estimates across groups and for both labels, but that this happens in the long-run and given access to sufficient samples. In particular, we note that

---

[3]While this experiment maintains the same mapping throughout, the mapping could be periodically revised.

for label 1 agents from $G_b$, as there are only 1080 samples in the dataset, although the bias initially decreases, the final estimate still differs from the true value. Fig. 4(c) verifies that this estimate would have been debiased in the long-run, had additional samples from the underlying population become available (i.e., as more such agents arrive).

`Active debiasing` **on the *FICO* dataset:** Fig. 4 also illustrates the performance of our algorithm on the *FICO* dataset [37, 16], and shows that it is successful in debiasing distribution estimates on both groups and on both labels.

# 6 Conclusion, Limitations, and Future Work

We proposed an `active debiasing` algorithm which recovers unbiased estimates of the underlying data distribution of agents interacting with it over time. We also analyzed the interplay of our proposed statistical/data debiasing effort with existing social/model debiasing efforts, shedding light on the potential alignments and conflicts between these two goals in fair algorithmic decision making. We further illustrated the performance of our proposed algorithm, and its interplay with fairness constraints, through numerical experiments on both synthetic and real-world datasets.

**The single-unknown parameter assumption.** Our work focuses on learning of a single unknown parameter (Assumption 1). Despite the commonality of this assumption in the multi-armed bandit learning literature, it also entails parametric knowledge of the underlying distribution with the other parameters such as variance or spread being known. We extend our algorithm to a Gaussian distribution with two unknown parameters in Appendix I. Extensions beyond this, especially those not requiring parametric assumptions on the underlying distributions, remain a main direction of future work.

**On one-dimensional features and threshold classifiers.** Our analytical results have been focused on one-dimensional feature data and threshold classifiers. These assumptions may not be too restrictive in some cases: the optimality of threshold classifiers has been established in the literature by, e.g., [8, Thm 3.2] and [36], as long as a multi-dimensional feature can be mapped to a properly defined scalar. Moreover, the recent advances in deep learning have helped enable this possibility: one can take the last layer outputs from a deep neural network and use it as the single dimensional representation. That said, any reduction of multi-dimensional features to a single-dimensional score may lead to some loss of information. In particular, our experiments have considered the use of our `active debiasing` algorithm on the *Adult* dataset with multi-dimensional features by first performing a dimension reduction to a single-dimensional score; we find that this reduction can lead to a $\sim 5\%$ loss in performance (see Appendix A for details). One potential solution to this is to adopt a mapping from high-dimensional features to scores that is revised repeatedly as the algorithm collects more data. Alternatively, one may envision a debiasing algorithm which targets its exploration towards collecting data on features that are believed to be highly biased; these remain as potential extensions of our algorithm.

**Potential social impacts.** More broadly, while our debiasing algorithm imposes fairness constraints on its exploitation decisions (see problem (1)), it does not consider fairness constraints in its *exploration* decisions. That means that our proposed algorithm could be disproportionate in the way it increases opportunities for qualified or unqualified agents in different groups during exploration. Imposing fairness rules on exploration decisions, as well as identifying algorithms that can improve the speed of debiasing of estimates on underrepresented populations, can be explored to address these potential social impacts, and remain as interesting directions of future work.

Additional discussions on limitations, extensions, and social impacts, are given in Appendix A.

## Acknowledgments and Disclosure of Funding

We sincerely thank the three reviewers and the area chair for their comments and feedback which helped improve our paper. We are also grateful for support from the National Science Foundation (NSF) program on Fairness in AI in collaboration with Amazon under Award No. IIS-2040800, the NSF under grant IIS-2143895, and Cisco Research. Any opinion, findings, and conclusions or recommendations expressed in this material are those of the authors and do not necessarily reflect the views of the NSF, Amazon, or Cisco.

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
