# OpenReview forum: "Adaptive Data Debiasing through Bounded Exploration"
_NeurIPS.cc/2022/Conference — NeurIPS 2022 Accept_

### Official Review · Reviewer_e1EM · 2022-07-11

**Rating:** 7
**Confidence:** 2
**Soundness:** 2 fair
**Presentation:** 2 fair
**Contribution:** 3 good

**Summary:**

The paper considers an algorithm to debias datasets through adaptive and bounded exploration for classification. In particular, the bias in the data is defined as inaccuracy in the estimation of a univariate distribution (assumed known to be fit). Their algorithm works by adaptively picking a lower bound corresponding to a function of the percentile of the estimated univariate distribution (with current parameters). Then given a decision threshold classification algorithm, a threshold is fitted to determine if samples / agents are accepted, where violation of the threshold can occur wrt to the lower bound calculated (with low probability). Data is then selected for: (1) retraining the classifier; and (2) updating the parameters of the distribution.The paper presents properties of a purely exploitation and purely exploitation algorithms, and further shows that their algorithm has a favorable properties in a mix. This leads to convergence of estimated parameters. Regret analysis is done on the classifiers learnt; and the analysis of how a fairness constraint can effect the speed of convergence is done. Experiments are presented over synthetic and real world datasets.

---

*Update*

I have updated my score to a 7. Similar to other reviewers, I would also highlight that additional clarity in the paper would improve the paper greatly.

One specific point on this is to make the types of fairness less ambiguous. In the discussion and in the author responses, language has been used for different types of fairness which is confusing. There are two (broad) types of (un)fairness being highlighted: (1) from data; versus (2) from prediction. The authors are focusing on the former. However, the language used to describe both is confusing. For instance, when describing prediction fairness, terms like "social bias/model bias/unfairness" are used, but data fairness is also "social" and "unfairness" is too broad of a term. In my opinion, this should simply be stated as "prediction fairness" or something similar. On the other hand, data fairness is labelled "statistical bias/data bias". Here, the "statistical bias" term could refer to predictive fairness as constraints for this are "statistical" in nature. Simply put, the language here needs to be made specific, especially so in fairness where there are many "flavors".

**Questions:**

Questions / Comments / Suggestions
1. Algorithm 1 is not especially clear in the psuedo-code (/ main-text description). In particular, it is unclear that "new data" is used to retrain the classifier (which I believe is the case from what I have seen via code / appendix). Furthermore, even in the appendix, algorithm 1, it does not specify what data the retraining is being done with. Also it seems to be incorrect? The way it is currently written seems to suggest that it should be looping over $ t < T $ (alongside the currently loop(?) over samples / batches).
2. In the main-text / proofs / definitions, "wlog" statements are given for what the parameters are specified as (mean, percentile). Following the logic of the paper it appears that if we can prove, \ie, Theorem 2, for a single type of statistic(?) of the distributions, then it will work for any arbitrary statistic / parameter. If true, I assume this is from assumption 1? It would help to clarify this step.
3. Furthermore, with this assumption, it appears that the exploration and exploitation baselines also depend on how the parameters are specified. This is not clear. In particular, Theorem 1 & 2 are wrt the distribution parameters. In Theorem 2 the mean is specified, but what about Theorem 1? Is it the median as discussed on line 181?
4. Theorem 4, the regret bound, has many undefined symbols in the main-text.
5. The figures in the main-text are confusing. They state some algorithms "successfully debiases data". However, from the information given in the main-text, there is no way to verify the claim. For instance, for Fig 1, (a) + (b), how can we tell that the mean estimate is correct, when we don't know the true value? The corresponding figures in the appendix does have this information in their title.
6. Statistical data bias being specified as the error in parameters is confusing. I initially though this criteria such as the statistical parity or representation rate of the data.

Minor
  - Line 213 "overestiamted"
  - Appendix Algorithm 1 does not initialize $t$

**Limitations:**

The primary limitations are stated.

**Strengths And Weaknesses:**

Strength
  - Experiments show recovery of shifted parameters.
  - Theoretic results provide convergence and regret analysis.

Weaknesses
  - Writing can be improved. There are aspects of the paper which are very unclear. (enumerated below).
  - Figures in the main text are confusing / missing information.
  - Algorithm / Pseudo-code are not clear.

---

> ### Author Response · Authors · 2022-08-02
> **Thank you for your valuable suggestions and comments. We address the questions in the comments below.**
>
> Thank you for your valuable questions, comments, and suggestions.
>
> **Major comment 1**: Thank you for your careful reading. The $\hat{\omega}^y_t$ is updated by "new data", but the classifier is trained by all collected samples over time. Sorry for any confusions made, we have added more clarifications in the pseudo-code in the revised draft.
>
> **Major comment 2**: You are absolutely correct that based on our Assumption 1 (single parameter unknown), Theorem 2 holds for any arbitrary statistic. Take a Gaussian distribution as an example; if we only assume the mean $\mu$ is unknown, then the distribution shape (how the distribution spreads out) is fixed. Based on the law of large number, as we collect i.i.d samples from distribution, we are able to find any arbitrary statistics (e.g., 30\%, 50\%, 70\% percentile). After that, we can add / subtract the distance from these percentile to median (50\%) to get the unknown parameter $\mu$ because the sd $\sigma$ is fixed, which means the distance from any statistics to median is a constant. We have added this discussion at the conclusion of the proof of the theorem.
>
> **Major comment 3**: Thank you for noting this. In line 180, we describe the general case where $\hat{\omega}^0_t$ is the $\alpha$-th percentile of $\hat{f}^0_t$. In line 181, we provide an example when $\alpha =50$, which is the median of the distribution; also, in Theorem 1 and 2, we use the mean wlog. We have clarified this in Theorem 1 in the revised draft. We note that under Assumption 1, we can map one-to-one from any $\alpha$-percentile to another, and therefore the choice of percentiles are without loss of generality.
>
> **Major comment 4**: Thank you for the detailed reading and noting this. In the revised draft, we have added more explanations and details for each symbol in the theorem statement.
>
> **Major comment 5**: Thanks for your careful reading. Due to the limited space, by adding the ground-truth values in the main-text Fig. 1 will make graphs even smaller. Hence, we do not include a title in the main-text, but we do include these ground-truth information in the appendix with larger figures. Sorry for any confusions made. We have added some clarifications in the revised draft.
>
> **Major comment 6**: Thank you for your comments on the notion of bias. We would like to first note that our proposed active debiasing algorithm can also be viewed as one that is only used for data collection. That is, this data collection algorithm can be separate from the prediction model that eventually takes the collected data as input to generate useful predictions for the policy maker.
>
> With this in mind, in terms of relation to unfairness/"bias", as noted in our introduction and as classified by prior work (e.g., Mehrabi et al. in our references), there are two general sources of algorithmic bias: (1) social bias/(prediction) model bias/fairness, and (2) statistical bias/data bias; we focus on the latter, recognizing that it has implications for the former. Specifically, if we start with biased training data (statistical bias), the trained (prediction) model will be different from the desired/true model. This could also mean that trained model is unfair, as studied and verified, e.g, by [1,2,3]. Therefore, our relation to the existing literature on social bias/unfairness is that unbiased/correct data is the first step towards fair AI. We also establish a connection (Proposition 1) between socially fair models and our algorithm for adaptively collecting statistically unbiased data.
>
> [1] Kallus, Nathan, and Angela Zhou. "Residual unfairness in fair machine learning from prejudiced data." International Conference on Machine Learning. PMLR, 2018.
>
> [2]  Wang, Jialu, Yang Liu, and Caleb Levy. "Fair classification with group-dependent label noise." Proceedings of the 2021 ACM conference on fairness, accountability, and transparency. 2021.
>
> [3] Zhu, Zhaowei, Tianyi Luo, and Yang Liu. "The rich get richer: Disparate impact of semi-supervised learning." ICLR 2022.
>
> Lastly, in our study of data bias, we have formalized data biases as the mismatch in the parameter of the estimated and true (joint) feature-label distributions $f^y(x)$ (Lines 138-139). We do not consider the specific source that has resulted in this mismatch; it could be any of the issues of data bias such as underrepresentation from a group, data shifts over time, incorrect/selective labeling in the past, feature measurement errors, etc. Our model of bias allows us to propose an algorithm for adaptively collecting an unbiased dataset while remaining agnostic to the original source of data bias.
>
> **Minor comments**: Thank you for your careful reading and comments. We have fixed typos and added the initialization for algorithm 1 in the appendix in the revised draft.

---

> > ### Comment · Reviewer_e1EM · 2022-08-08
> > **Thank you for the response**
> >
> > Thank you for the response. I will update the numeric score to a 6, with possible further change after discussion with other reviewers.
> > I am happy with the response given by the authors. I believe that the series of clarification given in the responses regarding Assumption 1 and its implications are definitely needed. I also appreciate the additional clarity given to the type of fairness / bias explored.
> >
> > ---
> > Also note: there is a missed typo in the revised version (and in the original version): Line 138 (of the revised), "mistmatch".

---

> ### Author Response · Authors · 2022-08-07
> **Response follow-up**
>
> Dear Reviewer e1EM,
>
> Once again, we appreciate your time devoted to reviewing this paper. We have provided responses to your comments and an updated submission. Could you please check whether they properly addressed your concern? Your feedback would be appreciated. Please kindly let us know in case there are other concerns--we hope we will have the opportunity to respond to them. Thank you very much!

---

### Official Review · Reviewer_ESAg · 2022-07-11

**Rating:** 6
**Confidence:** 3
**Soundness:** 3 good
**Presentation:** 2 fair
**Contribution:** 3 good

**Summary:**

The paper presents a method for "data debiasing" in a sequential-data setting with one feature (analysis extended to two in supplement). The algorithm conducts a form of bounded-random exploration in this one-dimensional setting, and can be used to perform learning under fairness constraints. The authors provide a theoretical analysis of the proposed algorithm, demonstrating convergence and analyzing the regret bound. The paper includes experiments on a simluated datasets and versions of two fairness datasets.

Overall, the paper is mostly well-written and notation clear (with some exceptions, noted below). I have three main concerns about the paper. The first is about the practical significance of the problem: this seems to be of limited real-world significance, and the authors do not identify a meaningful application where such a model and the described setting hold. Second, I have some concerns with the presentation of the work as "data debiasing", as their work is really about model debiasing in an active-sampling setting. Third, I find the real-world experiments somewhat unconvincing (see below).

## After response

See comments below.

**Questions:**

See above.

**Limitations:**

See above.

**Strengths And Weaknesses:**

#### Major Comments

* This work strikes me as inappropriately framed, in two ways. First, it seems to be about *model* debiasing, not data debiasing. For example, the authors' own definition of bias is a property of the model, not of the data (cf. L43-49; L125-129). Second, this definition of "bias" seems wholly unrelated to any fairness concerns and is effectively a measure of model error. The work would make sense as a fairness work if their measure of bias was, for example, related to disparities in this error over subgroups, but as-is, the proposed algorithm seems to simply be a one-dimensional active-learning method that could incorporate fairness (or any other) constraints.

* The entirety of the main text focuses on the case of one-dimensional data (with a reference to 2-D case in supplement). If this is the case, the proposed method seems to be of limited usefulness in most real-world classification scenarios, where many features are typically available. The authors' own somewhat contrived experiments on "real-world" datasets (Adult, FICO) which reduce these datasets to a single feature in order to apply their proposed method, seem to demonstrate the limited real-world usefulness. I understand that this may be an interesting theoretical case for the bandit literature, but the authors' claim to solve a meaningful fairness problem is hard to take at face value without any clear examples of such a scenario emerging in the real world.

* The real-world data experiments, in particular, are weak. The authors do not provide the ground-truth values in Fig. 4a/4b/4c/4d; only one panel compares the proposed method to baselines; and the synthetic data augmentation in 4(c) seems ad hoc and is not described. Most concerning, however, is the authors' distilling these richly-featured tabular datasets into a single feature for the purposes of their experiments, which amounts to discarding information in order to shoehorn in the proposed method. This is effectively ignoring hundreds of published results on these datasets in the fairness literature, which would perhaps be an appropriate baseline. The results are also missing any notion of classification error or loss, if I understand them correctly.

#### Minor Comments

* L131 "This type of assumption is common in the literature" - please provide relevant citations.

* Definition 1 could use more unpacking; some of the properties described as "intuitive" or otherwise are not immediately clear to me. A clearer description of the properties of LB_t would clarify the work considerably.

* The intuition in L304-312 is not clear; in particular the sentence in L306-L309 could use more unpacking. It currently reads, paradoxically, as "an increase in exploration makes the model more conservative at exploration".

#### Typos etc.


* I am confused why there is no "hat" on \theta, as there is on \omega. Isn't it the case that there exists a true \theta_a, \theta_b, which we are estimating via \hat{\theta}_a, \hat{\theta}_b, respectively?

* L154: in my opinion it would be clearer to retain the g subscript in the following analysis.

* "lowerbound" --> lower bound

* L178: "update the estimates to f..." --> update f

* L179: \theta_t^y is not an "unknown" parameter if we are updating it.

* L213: "overestiamted"

---

> ### Author Response · Authors · 2022-08-02
> **Thank you for your valuable suggestions and comments. We address the questions in the comments below.**
>
> Thank you for your valuable comments and suggestions.
>
> **Major comment 1**: Thank you for your comments on the notion of bias.
>
> We would like to first note that our proposed active debiasing algorithm shown in line 125-129 is one that is used for data collection (as opposed to prediction): our algorithm is estimating the data distribution and using it as a guideline to adaptively collect new data, in a way that leads to a statistically unbiased dataset. As such, this data collection algorithm can be separate from the algorithm (prediction model) that eventually takes the collected data as input to generate useful predictions for the policy maker. In other words, the active debiasing algorithm we propose is one that takes the initially observed training data as input and tries to estimate the correct distribution $P(X,Y)$ ($f^y(x)$ in our paper notation) of the data; this is in contrast to a prediction model takes $X$ as an input and outputs predictions of $Y$.
>
> In terms of relation to fairness, as noted in our introduction and as classified by prior work (e.g., Mehrabi et al. in our references), there are two general sources of algorithmic bias: (1) social bias/(prediction) model bias/unfairness, and (2) statistical bias/data bias. We focus on the latter, recognizing that it has implications on the former. Specifically, if we start with biased training data (statistical bias), the trained (prediction) model will be different from the desired/true model. This could also mean that trained model is unfair, as studied and verified, e.g, by [1,2,3]. Therefore, our relation to the existing literature on social bias/unfairness (as noted in our introduction and related work) is that unbiased/correct data is the first step towards fair AI. We also establish a connection (Proposition 1) between socially fair models and our algorithm for adaptively collecting statistically unbiased data.
>
> [1] Kallus, Nathan, and Angela Zhou. "Residual unfairness in fair machine learning from prejudiced data." International Conference on Machine Learning. PMLR, 2018.
>
> [2]  Wang, Jialu, Yang Liu, and Caleb Levy. "Fair classification with group-dependent label noise." Proceedings of the 2021 ACM conference on fairness, accountability, and transparency. 2021.
>
> [3] Zhu, Zhaowei, Tianyi Luo, and Yang Liu. "The rich get richer: Disparate impact of semi-supervised learning." ICLR 2022.
>
> Lastly, in our study of data bias, we have formalized data biases as the mismatch in the parameter of the estimated and true (joint) feature-label distributions $f^y(x)$ (Lines 138-139). We do not consider the specific source that has resulted in this mismatch; it could be any of the issues of data bias such as underrepresentation from a group, data shifts over time, incorrect/selective labeling in the past, feature measurement errors, etc. Our model of bias allows us to propose an algorithm for adaptively collecting an unbiased dataset while remaining agnostic to the original source of data bias.

---

> > ### Author Response · Authors · 2022-08-02
> > **(Cont.) Response**
> >
> > **Major comment 2**: Thank you for you comment on the importance of applicability of our work to multi-dimensional data. We hope to provide support for this below. Our current analytical work in the paper indeed discusses one-dimensional feature data and threshold classifiers, and our experiments also consider the algorithm used on data with multi-dimensional features by performing a dimension reduction (e.g., in our Adult dataset experiment; the FICO credit scores did not require such reduction).
> >
> > In terms of whether this dimension reduction will lead to limitations due to considerable information loss, we have run an additional set of experiments on the Adult dataset, with classifiers trained with and without performing dimension reduction. The experiments show only minimal loss in accuracy (<1\%), with results as follows:
> >
> > $\bullet$ For the classifier trained through logistic regression without performing dimension reduction: the overall accuracy is 78.44\%, and the accuracy for advantaged (41762 samples) and disadvantaged (7080 samples) group are 77.42\% and 85.27\%, respectively.
> >
> > $\bullet$ For the classifier trained through logistic regression after performing dimension reduction: The overall accuracy is 77.79\%, and the accuracy for the advantaged and disadvantaged group are 76.73\% and 84.04\%, respectively.
> >
> > Hence, such reduction and focus on threshold classifier might not seem as restrictive as it sounds: its optimality has been established in the literature by Corbett-Davies et al. [1, Thm 3.2] and Raab \& Liu [2] as long as a multi-dimensional $X$ can be mapped to a properly defined scalar. The recent advances in deep learning in fact has helped enable this possibility: for instance, one can take the last layer output from a deep neural network and use it as the single dimensional representation. As an example, one can collect a variety of information about a person's financial information and train a model to combine them into a single dimension $\in [0, 850]$, i.e., a credit score. Then the classification question of whether someone's loan application should be approved or not reduces to finding a threshold of the risk score to determine the decision.
> >
> > We have added the discussion above on single-dimensional data and threshold classifiers in our discussion section in Appendix A in our revised draft (lines 539-560). We will move a summary of this discussion to a conclusion section in our final draft as well (given the extra space).
> >
> > [1] Corbett-Davies, Sam, et al. "Algorithmic decision making and the cost of fairness." Proceedings of the 23rd acm sigkdd international conference on knowledge discovery and data mining. 2017.
> >
> > [2] Raab, Reilly, and Yang Liu. "Unintended selection: Persistent qualification rate disparities and interventions." Advances in Neural Information Processing Systems 34 (2021): 26053-26065.
> >
> > **Major comment 3**: Thanks for your careful reading. Due to the limited space, by adding the ground-truth values in the main-text Fig. 4 will make graphs even smaller. Hence, we have not included these as plot titles in the main-text, but we have included these ground-truth information in the appendix with larger figures (Fig. 10 caption). For the classification error or loss, we have included additional experiments in the appendix (Fig. 12) to compare the classification error with different depth of exploration, and a regret analysis for the loss. The ground-truth information for Fig. 12 is shown as follows: The initial biased distributions are Beta(2,3) and Beta(5,5) for label 1 and 0 respectively, and the true distributions are Beta(5,3) and Beta(3,5) respectively. Similarly, the ground-truth information for \emph{Adult} and \emph{FICO} dataset can be found in Fig. 10 and Fig. 11 caption. Sorry for any confusions made, we have added more clarifications in the revised draft. In terms of the impacts of dimensionality-reduction we have used in the Adult dataset, in the previous answer to "Major comment 2", we have provided an additional experiment results and discussions on why we believe the dimensionality reduction on the features and our focus on threshold classifiers is not too restrictive. We have also added this discussion in Appendix A in our revised draft.

---

> > > ### Author Response · Authors · 2022-08-02
> > > **(Cont.) Response**
> > >
> > > **Minor comment 1**: The single parameter unknown assumption is used, e.g., in the following works in the multi-armed bandit and fair learning literatures, which we will make sure to add to L131:
> > >
> > > [1] Slivkins, Aleksandrs. "Introduction to multi-armed bandits." Foundations and Trends® in Machine Learning 12.1-2 (2019): 1-286.
> > >
> > > [2] Patil, Vishakha, et al. "Achieving Fairness in the Stochastic Multi-Armed Bandit Problem." J. Mach. Learn. Res. 22 (2021): 174-1.
> > >
> > > [3] Lattimore, Tor, and Csaba Szepesvári. Bandit algorithms. Cambridge University Press, 2020.
> > >
> > > [4] Schumann, Candice, et al. "Group fairness in bandit arm selection." arXiv preprint arXiv:1912.03802 (2019).
> > >
> > > [5] Raab, Reilly, and Yang Liu. "Unintended selection: Persistent qualification rate disparities and interventions." Advances in Neural Information Processing Systems 34 (2021): 26053-26065.
> > >
> > > **Minor comment 2**: Thank you for noting this. In Definition 1, in more detail, we choose $LB_t$ such that $\hat{F}^0_t(\omega^0_t)-\hat{F}^0_t(LB_t)=\hat{F}^0_t(\theta_t)-\hat{F}^0_t(\omega^0_t)$; that is, such that $\omega^0_t$ is the median in the interval $(LB_t, \theta_t)$ based on the current estimate of the distribution $\hat{F}^0_t$ at the beginning of time $t$. Then, we update $\omega^0_t$ to $\omega^0_{t+1}$, the \emph{realized} median of the distribution between $(LB_t, \theta_t)$ based on the observed data during $[t, t+1)$. Once the underlying distribution is correctly estimated, (in expectation) we will observe the same number of samples between $(LB_t, \omega^0_t)$ and between $(\omega^0_t, \theta_t)$, and hence $\omega^0_t$ will no longer change. We will add this explanation after Definition 1 in the final draft (given the extra space).
> > >
> > > **Minor comment 3**: Thanks for noting this. In terms of the seemingly paradoxical statement: the first ``increase'' is stating the increase in the exploration \emph{threshold} $LB_t$ (as opposed to increase in data exploration/collection). If the decision threshold $\theta_t$ becomes smaller, by Definition 1, the corresponding lower bound $LB_t$ will become larger. As a result, our algorithm's exploration range $[LB_t, \theta_t]$ becomes narrower overall, and that is why our algorithm becomes more conservative at (data) exploration. We will make sure to work on rephrasing that discussion for clarity in the final draft.
> > >
> > > **Typos, etc 1**: Thank you for your careful reading. You are right that there are true $\theta_a, \theta_b$ (they never change w.r.t. time $t$), and we are estimating via $\theta_{a,t}, \theta_{b,t}$. Here, $\theta_{a,t}$ is the same as $\hat{\theta}_{a,t}$. We have modified our notations to make it consistent with other notations in the revised draft.
> > >
> > > **Remaining typos, etc**: Thank you for your careful reading and comments. We have fixed these typos in the revised draft.

---

> ### Author Response · Authors · 2022-08-07
> **Response follow-up**
>
> Dear Reviewer ESAg,
>
> Once again, we appreciate your time devoted to reviewing this paper. We have provided responses to your comments and an updated submission. Could you please check whether they properly addressed your concern? Your feedback would be appreciated. Please kindly let us know in case there are other concerns--we hope we will have the opportunity to respond to them. Thank you very much!

---

### Official Review · Reviewer_Hsp8 · 2022-07-13

**Rating:** 7
**Confidence:** 4
**Soundness:** 3 good
**Presentation:** 2 fair
**Contribution:** 3 good

**Summary:**

The work studies the problem of collecting new data to learn outcome distribution of sensitive groups. It analyses three schemes of data collection — (exploitation only) one in which decisions to give a positive decision to individuals are solely based on current classifier, and (exploration only and active debiasing) other two where the individuals who would not have been given a positive decision are given so randomly with or without some constraints. Theoretical analysing the cases of unimodal and univariate distributions, the work shows that the latter two schemes estimate the outcome distributions correctly. It also proves bounded regret in making correct decisions for the active debasing scheme. These observations are verified in experiments on two real datasets.

**Questions:**

Please address the following five points in the response.

1. Assumption 1 is not clear mainly because the parameter \omega is not explained. Similarly Definition 1 is introduced without giving the context for the formula in line 159-161.

2. Clarify whether the analysis is restricted to single dimensional data and threshold classifiers. Extensions to multiple dimensions is not discussed.

3. Clarify the term speed of debiasing in Proposition 1. The statement and the proof are not precise.

4. Related work can be made more comprehensive by discussing important papers on the three relevant parts of the problem - selective labelling, fair learning from imperfect data, and active learning of fair models. For example, see Lakkaraju et al. 2017
(https://dl.acm.org/doi/10.1145/3097983.3098066) and De-Arteaga et al. 2018 (https://arxiv.org/abs/1807.00905) for selective labeling, and Blum and Stangl 2019 (https://arxiv.org/abs/1912.01094) and Kallus and Zhou 2018 (https://arxiv.org/abs/1806.02887) for fairness learning from imperfect data. For active debiasing please see Abernethy et al. 2020 (https://arxiv.org/abs/2006.06879) and Noriega-Campero et al. 2019 (https://dl.acm.org/doi/10.1145/3306618.3314277).

5. For the first four results the problem seems to be more related to online mean estimation than fair learning — how to learn two parameters (which are means of some distribution) online with the difference being the constraint of bounded exploration. The final result Proposition 1 is interesting in terms of interplay between online data collection and putting fairness constraints. Because of this, I would suggest discussing differences from the related work on online mean estimation. Please motivate the unique aspects of the problem setting like how the bounded exploration makes the problem and the analysis harder.


Minor (no response is requested for these)

The presentation of the problem can be improved in the Introduction. It is a bit stylistic preference, but I would suggest adding a discussion of what has been done in the past work in Introduction itself and pointing out the shortcomings and improvements by the current work.
More importantly, the idea of bias in data is not clear. Does this refer to the training data being censored i.e. labels are available only for individuals with positive predictions? Or, the idea is more general? Lines 47-49 can be made more precise. For example, clarify if the mismatch between estimated and true label distribution refer to estimation error i.e. mismatch due to finite samples or the estimate is biased.

Please clarify whether the analysis depends on the form of the loss-miniminizing fair algorithm in Eq. (1). Does it matter if it is solved when the fairness constraints are defined with some slack (i.e. the difference need not be exactly zero) as done in many fair learning algorithms? Can the constraints be defined for differences in positive predictions (instead of the differences in thresholds) between the groups?

Define or provide a reference for alpha-stable distributions

unimodel -> unimodal

Add a conclusion section

**Limitations:**

Limitations of the work are not discussed in the main paper. Please discuss the limitations of the analysis for unimodal distributions and of the method in ignoring the feature information in data collection. Please discuss related work and use it to motivate the uniqueness of the problem setup. Some of the discussion from Appendix A can be moved to the main paper. Potential negative impacts are discussed.

**Strengths And Weaknesses:**

Strengths
- interesting and overlooked problem of how new data collection impacts fairness guarantees
- theoretical analysis poses reasonable questions on learnability in this setting and finds intuitive answers
- generally well written paper

Weaknesses
- some terms are not made precise like bias or debiasing
- most relevant work are not discussed in the main paper or Appendix
- limitations of the assumptions like unimodal distributions and of the method like not looking at the features are not discussed

Overall I like the research direction of studying the interplay between data collection and fairness when starting from imperfect data. Proposition 1 is a good step towards understanding this interplay.


---
## After the response

The response addresses my queries adequately. Hence I have updated my score to 7, Accept. The paper explores interesting ideas on controlled data collection to correct biased estimates and the effect of putting fairness constraints during collection. There are limitations like the restriction to single dimensional data. Overall, the work makes a good contribution.

I appreciate the thorough review of related work in the response. Please include the discussion into the main paper differentiating the problem setting and analysis from past work. Clarity of presentation needs to be improved as noted by the other reviewers including clarifying the problem and notation. I would suggest addressing the restriction to single-dimensional data by pointing to ways in which the insights can be transferred to the multi-dimensional case. It is unreasonable to expect that dimension reduction would be as lossless as shown in the Adult Income experiment (also, I could find the exact procedure used for dimension reduction, so the result is a bit surprising if it is completely unsupervised).

---

> ### Author Response · Authors · 2022-08-02
> **Thank you for your valuable suggestions and comments. We address the questions in the comments below.**
>
> Thank you for your valuable comments and suggestions.
>
> **Major comment 1**: Thank you for noting these. We let $\hat{\omega}^y_t$ be the $\alpha$-th percentile of $\hat{f}^y_t$ in our algorithm. (This has now been clarified in lines 135-136 after Assumption 1 in the revised draft.) As a simple instance, when $\alpha = 50$ and $\hat{f}^y_t$ is Gaussian, this $\hat{\omega}^y_t$ is the median of $\hat{f}^y_t$. Assumption 1 then states that the firms will update the estimated distribution $\hat{f}^y_t$ by updating its mean/median $\hat{\omega}^y_t$ only (without updating its variance). We have stated the assumption more generally, as our choice of an $\alpha$-percentile is wlog. For instance, the unknown parameter could be the rate parameter $\lambda$ of an exponential distribution, or the $\beta$ of a Beta distribution; such unknown parameters have a one-to-one mapping to the distribution's $\alpha$-percentiles under Assumption 1, and therefore our algorithm can be used to update such unknown parameters as well by performing the appropriate mapping.
>
> In Definition 1, in more detail, we choose $LB_t$ such that $\hat{F}^0_t(\hat{\omega}^0_t)-\hat{F}^0_t(LB_t)=\hat{F}^0_t(\theta_t)-\hat{F}^0_t(\hat{\omega}^0_t)$; that is, such that $\hat{\omega}^0_t$ is the median in the interval $(LB_t, \theta_t)$ based on the current estimate of the distribution $\hat{F}^0_t$ at the beginning of time $t$. Then, we update $\hat{\omega}^0_t$ to $\hat{\omega}^0_{t+1}$, the \emph{realized} median of the distribution between $(LB_t, \theta_t)$ based on the observed data during $[t, t+1)$. Once the underlying distribution is correctly estimated, (in expectation) we will observe the same number of samples between $(LB_t, \omega^0_t)$ and between $(\omega^0_t, \theta_t)$, and hence $\omega^0_t$ will no longer change. Thanks for pointing this out; we will add this explanation after Definition 1 in the final draft (given the extra space).
>
> **Major comment 2**: Our current analytical work indeed focuses on one-dimensional feature data and threshold classifiers, and our experiments also consider the algorithm used on data with multi-dimensional features by performing a dimension reduction (e.g., in our Adult dataset experiment).
>
> In terms of whether there is a considerable information loss due to such dimension reduction, we have run an additional set of experiments on the Adult dataset, with classifiers trained with and without performing dimension reduction. The experiments show only minimal (<1%) loss in accuracy, with results as follows:
>
> $\bullet$ For the classifier trained through logistic regression without performing dimension reduction: the overall accuracy is 78.44\%, and the accuracy for advantaged (41762 samples) and disadvantaged (7080 samples) group are 77.42\% and 85.27\%, respectively.
>
> $\bullet$ For the classifier trained through logistic regression after performing dimension reduction: The overall accuracy is 77.79\%, and the accuracy for the advantaged and disadvantaged group are 76.73\% and 84.04\%, respectively.
>
> In addition, the focus on a threshold classifier might not seem as restrictive as it sounds: its optimality has been established in the literature by Corbett-Davies et al. [1, Thm 3.2] and Raab \& Liu [2] as long as a multi-diemnsional $X$ can be mapped to a properly defined scalar. The recent advances in deep learning in fact has helped enable this possibility: for instance, one can take the last layer outputs from a deep neural network and use it as the single dimensional representation. As another example, one can collect a variety of information about a person's financial information and train a model to combine them into a single dimension $\in [0, 850]$, i.e., a credit score. Then the classification question of whether someone's loan application should be approved or not reduces to finding a threshold of the risk score to determine the decision.
>
> We have added the discussion above on single-dimensional data and threshold classifiers in our discussion section in Appendix A in our revised draft (lines 539-560). We will move a summary of this discussion to a conclusion section in our final draft as you have also suggested (given the extra space).
>
> [1] Corbett-Davies, Sam, et al. "Algorithmic decision making and the cost of fairness." Proceedings of the 23rd acm sigkdd international conference on knowledge discovery and data mining. 2017.
>
> [2] Raab, Reilly, and Yang Liu. "Unintended selection: Persistent qualification rate disparities and interventions." Advances in Neural Information Processing Systems 34 (2021): 26053-26065.

---

> > ### Author Response · Authors · 2022-08-02
> > **(Cont.) Response**
> >
> > **Major comment 3**: Thank you for noting this. We have added the missing details to the beginning of the proof of the proposition, and will update the proposition statement in our final draft as well. We compare the speed of debiasing based on the error in the parameter estimate,  $\mathbb{E}[|\hat{\omega}^y_t-\omega^y|]$. Given a fixed $t$, the algorithm for which this error is larger has a lower speed of debiasing. In words, the slower algorithm needs to wait for \emph{more} arriving samples before it can reach the same parameter estimation error as a faster algorithm. Alternatively, the speed of debiasing comparison can be in terms of the number of arriving samples needed in order for the estimated parameter $\hat{\omega}^y$ to be within a given distance of the true parameter ${\omega}^y$, in expectation.
> >
> > To corroborate that Proposition 1 aligns with this definition, for instance, when a group is over-selected under a fairness constraint, the fairness-constrained threshold $\theta^F_{g,t}$ will be lower than the unconstrained threshold $\theta^U_{g,t}$. Therefore, the exploration range will be narrower, which means by adding a fairness constraint, the algorithm needs to wait and collect more samples (takes longer time) before it manages to collect sufficient data to accurately update the unknown distribution parameter, and hence, it has a slower debiasing speed.

---

> > > ### Author Response · Authors · 2022-08-02
> > > **(Cont.) Response**
> > >
> > > **Major comment 4**: Thank you for pointing out these important papers! We will make sure to include them in our reference list and related work in the final draft. Below, we provide a discussion of each paper and how our work relates to them:
> > >
> > > $\bullet$ From the selective labeling perspective (censored feedback in our paper): Lakkaraju et al. [1] address the problem of evaluating the performance of predictive model under the selective labeling problem. They propose a contraction technique to compare the performance of the predictive model and human judge while they are forced to have the same acceptance rate. Our work is similar in our focus on selective labeling, but we propose a bounded exploration technique to remove the difference between the estimated and true distributions, and avoid sample selection bias during data collection procedure. We also consider the cost of exploration and fairness consideration. In contrast, our work is more closely related to De-Arteaga et al. [2]. We both study the problems arising due to selective labeling. Similar to us, [2] proposes a data augmentation scheme by adding more samples that would be more likely rejected (we refer to this as exploration) to correct the sample selection bias. Their proposed data augmentation technique is similar to our bounded exploration, but it differs in its selection of samples in that it adds samples that would be more likely to be rejected.
> > >
> > > $\bullet$ From the fair learning from imperfect data perspective: We have discussed the paper from Blum and Stangl in the Appendix B. The second referred paper, Kallus and Zhou [3], shows that residual unfairness remains even after the adjustment for fairness when policies are learned from a biased dataset. They propose a re-weighting technique (similarly, re-weighing ideas are explored in [Blum and Stangl] and [Jiang and Nachum]) to solve the residual unfairness issue while accounting for the censoring/adaptive sampling bias. In contrast, we use bounded exploration to remove the mismatch between an incorrectly estimated and the true data distributions through additional data collection over time. We further consider the effects of fairness interventions as orthogonal to our proposed data collection procedure.
> > >
> > > $\bullet$ From the active learning perspective: We have some discussions about the active learning literature in Appendix B, and will augment them with the suggested papers.  Abernethy et al. [4] propose an active sampling and re-weighting technique by sampling from the worst off group at each step. Their goal is to build a computationally efficient algorithm with strong convergence guarantees to improve the performance on the disadvantage (highest loss) group while satisfying the notion of min-max fairness. Noriega-Campero et al. [5] propose an adaptive fairness approach, which adaptively acquires additional information according to the needs of different groups or individuals given information budgets, to achieve fair classification. Similar to the approaches of these papers, we also compensate for adaptive sampling bias through exploration (by admitting individuals who would otherwise be rejected). In contrast, we start with a biased dataset, and we primarily focus on recovering the true distribution by bounded exploration, accounting for the cost of exploration, avoiding the adaptive sampling bias, and consider fairness issues as orthogonal to our data collection procedure (and as such, can apply our procedure to debiasing the estimates on a single group).
> > >
> > > [1] Lakkaraju, Himabindu, et al. "The selective labels problem: Evaluating algorithmic predictions in the presence of unobservables." Proceedings of the 23rd ACM SIGKDD International Conference on Knowledge Discovery and Data Mining. 2017.
> > >
> > > [2] De-Arteaga, Maria, Artur Dubrawski, and Alexandra Chouldechova. "Learning under selective labels in the presence of expert consistency." arXiv preprint arXiv:1807.00905 (2018).
> > >
> > > [3] Kallus, Nathan, and Angela Zhou. "Residual unfairness in fair machine learning from prejudiced data." International Conference on Machine Learning. PMLR, 2018.
> > >
> > > [4] Abernethy, Jacob, et al. "Active sampling for min-max fairness." arXiv preprint arXiv:2006.06879 (2020).
> > >
> > > [5] Noriega-Campero, Alejandro, et al. "Active fairness in algorithmic decision making." Proceedings of the 2019 AAAI/ACM Conference on AI, Ethics, and Society. 2019.

---

> > > > ### Author Response · Authors · 2022-08-02
> > > > **(Cont.) Response**
> > > >
> > > > **Major comment 5**: Thank you also for pointing to the relation to the literature on online mean estimation. We will conduct and add a review of this literature in our related work section in the final draft. In summary, compared to this literature, the main technical challenges of our proposed bounded exploration in online mean estimation is that it involves evaluating the behavior of statistical estimates based on data collected from a truncated distribution with \emph{time-varying} truncation. More specifically, our data collection interval is bounded and truncated (which has been considered in some prior work on distribution/mean estimation as well, e.g., Lai and Ying [1991]) but our exploration interval $[LB_t, \infty)$ is itself adaptive (which we believe is the main new aspect) and is what has motivated our analysis in a finite sample regime in Theorem 3. As you have noted, our focus on the interplay of fairness constraints with online estimation efforts (Proposition 1) is also new compared to this existing literature, and we will make sure to emphasize that as well.
> > > >
> > > > [1] Lai, Tze Leung, and Zhiliang Ying. "Estimating a distribution function with truncated and censored data." The Annals of Statistics (1991): 417-442.
> > > >
> > > > **Minor comments**: Thank you for your careful reading and comments. We have added some clarifications and fixed the noted typos in the revised draft, and hope to add the additional clarifications and a conclusion section to our final draft.

---

> ### Author Response · Authors · 2022-08-07
> **Response follow-up**
>
> Dear Reviewer Hsp8,
>
> Once again, we appreciate your time devoted to reviewing this paper. We have provided responses to your comments and an updated submission. Could you please check whether they properly addressed your concern? Your feedback would be appreciated. Please kindly let us know in case there are other concerns--we hope we will have the opportunity to respond to them. Thank you very much!

---

> ### Comment · Reviewer_Hsp8 · 2022-08-07
> **After the response**
>
> Thank you for responding to the queries. The response addresses my queries adequately. Hence I have updated my score to 7, Accept. The paper explores interesting ideas on controlled data collection to correct biased estimates and the effect of putting fairness constraints during collection. There are limitations like the restriction to single dimensional data. Overall, the work makes a good contribution.
>
> I appreciate the thorough review of related work in the response. Please include the discussion into the main paper differentiating the problem setting and analysis from past work. Clarity of presentation needs to be improved as noted by the other reviewers including clarifying the problem and notation. I would suggest addressing the restriction to single-dimensional data by pointing to ways in which the insights can be transferred to the multi-dimensional case. It is unreasonable to expect that dimension reduction would be as lossless as shown in the Adult Income experiment (also, I could find the exact procedure used for dimension reduction, so the result is a bit surprising if it is completely unsupervised).

---

### Author Response · Authors · 2022-08-02
**Summary of our response**

We thank all the reviewers for their comments and valuable feedback. We have made the following updates following the reviews to further improve our work.

$\bullet$ 1: We discuss the implications of our reduction from  multi-dimensional data to a single-dimensional representation, in response to comments from reviewer Hsp8 and reviewer ESAg. In particular, in Appendix A, we have added discussion and results from an additional experiment showing only a small difference in a classifier's accuracy with and without performing dimension reduction on the Adult dataset.

$\bullet$ 2: We unpack our Definition 1 to give more explanations on how the lower bound is derived, following the suggestions from reviewer Hsp8 and reviewer ESAg.

$\bullet$ 3: We provide additional explanations on the unknown parameter $\hat{\omega}^y_t$  (the $\alpha$-th percentile of $\hat{f}^y_t$) in our algorithm throughout the paper, according to the comments from reviewer Hsp8 and reviewer e1EM.

$\bullet$ 4: We discuss the relation of our work to important papers referred by reviewer Hsp8 in the fields of selective labeling, fair learning, and active learning, and the relation to the online mean estimation literature.

$\bullet$ 5: We emphasize the differences between (1) social bias/(prediction) model bias/unfairness and (2) statistical bias/data bias, according to the comments from reviewer ESAg and reviewer e1EM. We emphasize that our focus is on the latter data bias issue, which has direct implications on the former algorithmic unfairness issue (we have added some citations in this regard).

$\bullet$ 6:We discuss the reason why there is no ground-truth information in the main-text, according to the comments from reviewer ESAg and reviewer e1EM. Due to the page limit, those important information were included primarily in the Appendix.

$\bullet$ 7: We add some citations regarding to our Assumption 1 (single parameter unknown assumption), following the suggestions from reviewer ESAg.

$\bullet$ 8: We formalize the term speed of debiasing according to the comment from reviewer Hsp8.

$\bullet$ 9: We add explanations and details for notation missing from Theorem 4 following the comment from reviewer e1EM.

Several of the updates have been made in the revised draft (the location of these revisions have been noted in the individual responses). If our manuscript is accepted, the additional content introduced and summarized above (including discussions about dimensionality reduction, unpacking of Definition 1, and additional related work) will be merged into the main text given the extra page limit for the camera-ready version.

---

### Meta-Review · Area_Chair_xthd · 2022-08-26

**Recommendation:** Accept
**Confidence:** Certain

**Metareview:**

This paper has seen a lot of discussion between reviewers and authors. Reviewers are fairly positive after the discussion/rebuttal phase and there have been significant score revisions upwards.

Few concerns that were highlighted during rebuttal/discussion phase are:
  1) Multiple reviewers have pointed out that amongst two sources of bias - data bias and model bias - the authors focus on assembling a dataset to avoid the first type of bias. It has been pointed out using terms like "social bias, unfairness" and "statistical bias" are very misleading . I strongly suggest the authors to better revise the paper according to reviewer comments using more precise terminology -data bias and/or model bias. Clarity has been a concern uniformly shared amongst all reviewers.

2) The authors principally reduce the data to a single dimension using dimension reduction techniques and use thresholded classifier. Authors responded to this concern saying -effective feature learning in general amounts to that and there are optimal data dimension reduction techniques. Further authors also experimentally demonstrate that losses in accuracy is not much due to these techniques.

In summary, concerns 1 and 2 are not severe enough (as acknowledged by reviewers raising scores) but important to keep in mind while preparing the camera ready.

**Award:**

No

---

### Decision · Program_Chairs · 2022-09-14

Accept